# BPKEM: A biometric-based private key encryption and management framework for blockchain

**Hao Cai, Han Li, Jianlong Xu\*, Linfeng Li, Yue Zhang**

Department of Computer Science, Shantou University, Shantou, China

\* xujianlong@stu.edu.cn

## Abstract

The fundamental technology behind bitcoin, known as blockchain, has been studied and used in a variety of industries especially in finance. The security of blockchain is extremely important as it will affects the assets of the clients as well as it is the lifeline feature of the entire system that needs to be guaranteed. Currently, there is a lack of a methodical approach to guarantee the security and dependability of the private key during its whole life. Furthermore, there is no quick, easy, or secure way to create the encryption key. A biometric-based private key encryption and management framework (BPKEM) for blockchain is proposed not only to solve the private key lifecycle manag- ement problem, but also it maintains compatibility with existing blockchain systems. For the problem of private key encryption, a biometric-based stable key generation method is proposed. By using the relative invariance between facial and fingerprint feature points, this measure can convert feature points into stable and distinguishable descriptors, then using a reusable fuzzy extractor to create a stable key. The correct- ness and efficiency of the newly proposed biometric-based blockchain encryption tech- nique in this paper has been validated in the experiments.

## 1 Introduction

To meet security requirements and establish ownership, asymmetric encryption is a cryptography method integrated into the blockchain. The public key and the private key are two distinct keys that are commonly utilized during encryption and decryption. The public key is the number produced by the private key through the elliptic curve encryption algorithm (ECC), which is used to generate the address used in bitcoin transactions. Additionally, the production of the public key is irreversible, making it impossible to flip the private key using the public key. Participants in a blockchain system hold various keys based on their position. The public key is made available to the transaction verifier, who uses it to confirm the legitimacy of the transaction as well as the identity of the transaction publisher. The private key must be kept confidential by the user themselves due to the decentralized nature of the blockchain system, and the user uses the private key to digitally sign transactions in order to manage its assets. Due to the immutability of the blockchain, the loss of private keys due to inappropriate use and storage

**Data Availability Statement:** All relevant data are within the paper.

**Funding:** Funding: This research was financially supported by 2021 Guangdong province special fund for science and technology project, Grant

Number: STKJ2021201, STKJ202209017;
Research on Food Production and Marketing
traceability Software system based on Blockchain,
Grant Number: STKJ2021011; Guangdong
Provincial Science and Technology Plan Project,
Grant Number: STKJ2021011; 2020 Li Ka Shing
Foundation Cross-Disciplinary Research Grant,
Grant Number: 2020LKSFG08D; Guangdong basic
and applied basic research fund project, Grant
Number: 2021A1515012527; Free application
project of Guangdong Natural Science Foundation,
Grant Number: 2018A030313438; Special projects
in key fields of colleges and universities in
Guangdong Province, Grant Number:
2020ZDZX3073,2022ZDZX1008. The funders had
no role in study design, data collection and
analysis, decision to publish, or preparation of the
manuscript.

**Competing interests:** The authors have declared
that no competing interests exist.

would result in a significant loss of benefits for blockchain users. The security of the private key has a direct impact on the security of all the user's assets and the system as a whole. Therefore, how to secure the security of the private key in the blockchain system is a highly important study topic.

There are many threats to private key security today. First of all, there are still great hidden dangers in the security of the generation of public and private key pairs. Since quantum computing poses a great threat to the security of blockchain cryptographic algorithms, the generation of public and private keys based on elliptic curves and digital signatures cannot resist quantum computing [1]. Therefore, it is necessary to introduce a cryptographic method that is resistant to quantum computing to encrypt the private key. Traditional keys are typically generated randomly by the system or chosen by the user, but the system's randomly chosen keys are forgettable and the user-chosen keys are susceptible to dictionary attacks and exhaustive search attacks. Biometric features are unique, difficult to forge, and provide privacy. Based on this, biometric keys are difficult to decipher, portable, and non-repudiating. Second, there are no efficient management solutions for the secure storage and recovery of private keys [2]. The current management mechanisms for private keys mainly include human brain memory, local storage, offline storage, escrow wallets, threshold wallets and other methods. Because the private key is long and irregular, and once any bit is forgotten, it will make the private key unavailable, using the human brain to remember the private key is a high-risk and difficult challenge. Local storage faces risks such as malicious reads and damage to physical devices; offline storage still needs to be connected to the Internet when used, and malware attacks cannot be completely avoided; hosted wallets undermine the principle of blockchain decentralization, and there are problems such as central node trust problems and hosting servers being attacked. Threshold wallet distributes private keys across multiple devices using threshold secret sharing technology, which is difficult to design, complex, and not scalable.

Therefore, it is of great practical significance to study the use of biometric cryptography to encrypt private keys, and to explore an efficient and systematic solution to ensure the security of private keys from generation, encryption and storage to recovery.

For the above problems, we propose a biometric-based stable key generation (BKG) method to encrypt the blockchain's private key. BKG mainly extracts stable and distinguishable descriptors through facial features and fingerprint features to further encrypt the blockchain's private key. Additionally, a thorough and organized plan is put out based on BKG to guarantee the security of the entire process of creating private keys, encrypting them, and recovering them.

The contributions of this work are as follows:

1. To address an efficient management solution for secure storage and recovery of private keys, we propose A comprehensive management solution for the blockchain's private key system is developed based on BKG. For the purpose of evaluating the effectiveness of the plan, tests were done on the blockchain.

2. For the problem of safe and convenient encryption of blockchain private keys, we propose a method for creating distinguishing descriptors that is reliable, accurate, and based on several biometric factors. It can effectively screen fingerprint and facial biometrics, as the majority of a person's biometric characteristics vary throughout time. The problem of single biometric features being easily cracked is effectively resolved, and the error of feature descriptors is decreased, thanks to the usage of feature points in the generation of descriptors. Reusable fuzzy extractors are also used in key creation to ensure the security of user biometric features.

The structure of this paper is as follows: Section 2 presents related work. Section 3 introduces some preliminaries. The fundamental architecture, a biometrics-based private key encryption technique, and a blockchain private key security management scheme are all thoroughly introduced in Section 4. Section 5 shows the security and performance analysis about the scheme. The method and scheme are then experimentally validated in Section 6. Finally, Section 7 provides conclusions.

## 2 Related work

There is a burgeoning body of research on blockchain private key encryption. Presently, widely researched encryption technologies comprise of searchable encryption, homomorphic encryption, and biometric encryption. In this manuscript, we will examine these encryption methods and their strengths and limitations. On Searchable Encryption, in 2017, Li et al. [3] proposed a blockchain-based symmetric searchable encryption scheme that can complete multi-keyword searches, allowing users to obtain search results automatically without verification [4]. Yan et al. [5] amalgamated symmetric searchable encryption with attribute encryption to realize one-to-many searchable encryption and implemented fine-grained access control using a ciphertext policy attribute encryption mechanism for shared keys. However, these schemes are afflicted with issues of high complexity [4] and low efficiency [5]. On Homomorphic Encryption, Zheng et al. [6] utilized homomorphic encryption and threshold Paillier cryptosystem to encrypt private keys, facilitating shared data trading, and transaction information protection using (p,t)-threshold Pallier cryptosystem. Wang et al. [7] proposed a copyright blockchain privacy protection scheme based on lightweight homomorphic encryption and zero-knowledge proof. Nonetheless, existing homomorphic encryption algorithms still suffer from low efficiency [7] and excessive key size [6] and ciphertext explosion. Biometric encryption is one of the recently extensively studied techniques, Zhu et al. [8] proposed an efficient user login scheme for biometric authentication of any domain server based on blockchain nodes. Negin Hamian et al. [9] proposed a blockchain-based re-registration scheme for a biometric-based authentication system. The scheme employs Shamir's secret sharing, ElGamal encryption, and Schnorr's digital signature to calculate the secret biometric share. However, the description of the biometric application is too rudimentary. Aydar et al. [2] used fingerprints as biometrics to encrypt blockchain private keys. Carmen Bisogni et al. [10] proposed a blockchain private key encryption scheme based on face biometrics, which employs CNN to encode face features and then fuses them with RSA keys. Nevertheless, this scheme has relatively high lighting and background requirements. Bao et al. [11] proposed a novel identity authentication scheme combining fingerprint features and blockchain. Nevertheless, the efficiency of the fuzzy extractor employed is relatively low, and there is still room for improvement in security. To conclude, while there have been numerous strides in blockchain private key encryption, there are still challenges to surmount. Searchable encryption is beset with complexity and efficiency issues, homomorphic encryption encounters challenges with efficiency and key size, and biometric encryption necessitates improvement in its applications and security. Further research is indispensable to devise more efficient and secure blockchain private key encryption methods.

In addition to encryption, the storage and recovery of blockchain private keys is an essential guarantee for private key security. The private key storage stage includes various methods such as local storage, account custody [12, 13], offline storage [14, 15], cloud storage, encrypted wallet protection schemes. Although some researchers have proposed several methods to encrypt and store private keys, limitations still exist. For instance, Lusetti et al. [12] used symmetric and asymmetric methods to encrypt medical image files taken by forensics and uploaded them to an online platform and stored them in the blockchain. However, this method has some

limitations. Xiao et al. [14] proposed a multi-signature scheme based on Gamma signature. However, if the multi-signature is tampered or forged, the nodes in the tree need to verify part of the response top-down to find the malicious signer, which increases the running cost. Maria et al. [16] proposed a blockchain-based anonymous authentication scheme for providing secure communication in VANETs. In the proposed scheme, a Merkle Hash Tree (MHT) is used to understand the real-time authentication record, and blockchain-assisted successful V2R anonymous handover authentication is proposed when the vehicle user becomes a valid member of the in-vehicle network through initial anonymous authentication. Rapid re-authentication of the vehicle is achieved through secure verification code transmission between successive RSUs. Fan et al. [17] proposes a safe and reliable data transmission scheme based on the blockchain-based Internet of Things environment. In the scheme, the key generation center generates a private key for each base station and IoT node. After the IoT node enters the Internet of Things for the first time, it and the base station perform identity signature mutual verification. After successful verification, the base station generates a negotiated key, encrypts the symmetric key, and sends it to the node. The node then uses the inherent key and symmetric key encryption information to send to the base station, and the information is uploaded to the blockchain after decoding. In the private key recovery stage, Xiong et al. [18] proposed a blockchain key protection scheme based on secret sharing by introducing a private key distribution method to recover lost private keys. Li et al. [19] proposed a secret sharing scheme based on double-threshold key protection to recover the private key in 2021. However, this method has a high computational overhead, and the secure transmission of shadow sharing between blockchain nodes cannot be guaranteed. Many scholars have conducted in-depth research on a certain part of them, but there is no comprehensive and effective security solution for the generation, encryption, storage, and recovery of private keys.

In conclusion, private key encryption and management is a crucial factor in ensuring the security of blockchain-based systems. While various methods exist, they don't provide a comprehensive and effective security solution for the entire process of private key generation, encryption, storage, and recovery. Considering all the above issues, we propose a biometric-based blockchain private key encryption and management framework. Our framework achieves an appropriate trade-off between security and efficiency, realizing a proper balance between security and efficiency, it can be a significant step forward in the field of blockchain security.

## 3 Preliminaries

### 3.1 Fuzzy extractor

Fuzzy extractor [3] is a cryptographic technique used to extract secret keys from imperfect biometric data. Traditional biometric technologies, such as fingerprint recognition and facial recognition, usually only provide limited usable information and thus cannot be directly used as key material. Fuzzy extractor solve this problem by converting biometric data into entropy-density security keys. Fuzzy Extractor can convert a random source with a certain amount of noise into a uniformly random and accurately reproducible key. As shown in Fig 1, fuzzy extractor includes two parts: generation algorithm and regeneration algorithm, which can be expressed as Eq (1):

$$FE = (Gen, Rep) \tag{1}$$

In Fig 1, *Gen* means the generative algorithm, the generation algorithm *Gen* takes the input string *w* (one sample of the random source), and outputs a random value *R* and public

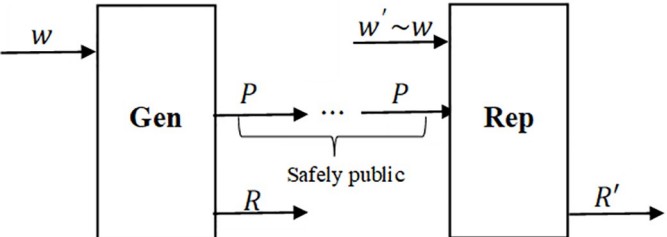

**Fig 1. The fuzzy extractor scheme.**

auxiliary data *P*. *Rep* means the regeneration algorithm, the regeneration algorithm *Rep* takes as input *w′* (another sample of the random source) and public auxiliary data *P*, and outputs the string *R′*. The fuzzy extractor generates the same random value *R* when the Hamming distance of the two samples is close enough, which means *R* = *R′*. The security requirement of the fuzzy extractor is that *R* is uniformly random if the random source has enough entropy.

## 3.2 Chinese remainder theorem

In mathematics, the Chinese Remainder Theorem (CRT) states that if one knows the remainders of the Euclidean division of an integer n by several integers, then one can uniquely determine the remainder of the division of n by the product of these integers, under the condition that the divisors are pairwise coprime (no two divisors share a common factor other than 1). CRT is a theorem that gives a unique solution to simultaneous linear congruences with prime moduli. Several versions of the Chinese remainder theorem have been proposed. The next one is called the general Chinese remainder theorem [20]: $m_1, m_2, \cdots, m_k$ are pairwise mutually prime positive integers that form the modular set $\beta = \{m_1, m_2, \cdots, m_k\}$, $M = \prod_{i=1}^{k} m_i$. When $X < M$, $X$ can be represented by unique set $\varphi = \{a_1, a_2, \cdots, a_k\}$, $X \equiv a_i (\bmod\ m_i)$, $(i = 1, 2, \cdots, k)$. When $\beta$ and $\varphi$ are known, $X$ can be recovered,

$$X \equiv \left( \frac{M}{m_1} e_1 a_1 + \frac{M}{m_2} e_2 a_2 + \cdots + \frac{M}{m_k} e_k a_k \right) (\bmod\ M), \text{ with } \frac{M}{m_i} e_i = 1 (\bmod\ m_i), (i = 1, 2, \cdots, k).$$

## 3.3 Learning with errors

Elliptic curve-based algorithms have been broken by quantum computing, and the use of post-quantum cryptography with resistance to quantum computing is necessary to resist quantum computing attacks. The problem of Learning with errors (LWE) [21], proposed by Regev, is recognized as having resistance to quantum properties and is necessary for post-quantum cryptography.

In linear algebra, a linear space can be described by finding a set of bases representing this space. if a set of bases of a linear space is known, then any vector in this space can be decomposed into a linear combination of this set of basis vectors. If a constraint is placed on this linear space, where the coefficients of all linear combinations must be integers, then the vectors generated by varying the coefficients of the linear combinations can only form a dense, lattice-like discrete set, as shown in Fig 2.

Generating a collection of spaces based on discrete basis vectors is known as an Integer Lattice. Since it is not possible to use linear combinations in the Integer Lattice to represent the exact vector we want, it is quite feasible to find a vector *v* that is closest to the vector *v′* that we want to represent and ensure that *v′* is exactly within the range that this Integer Lattice can represent. Then we need to find a set of integer coefficients to make *v′* the closest distance to the

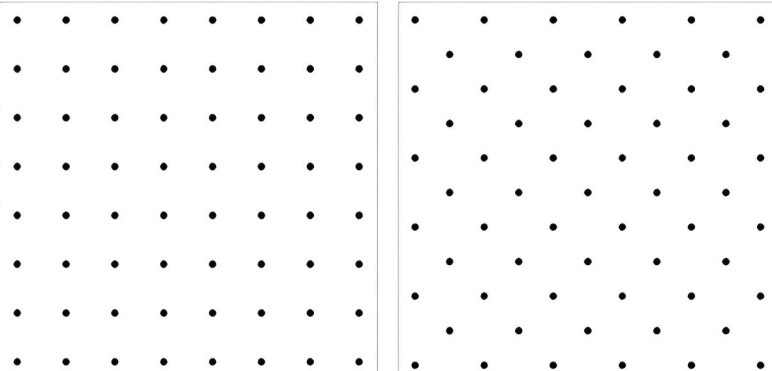

**Fig 2. Integer lattice, generating space sets based on discrete basis vectors.**

target vector in this integer lattice. This type of problem of approximating a target vector in a discrete linear set is collectively known as the Closest Vector Problem (CVP). The Learning With Errors (LWE) problem is derived from this problem. A matrix $A$ is known, and a vector $\hat{b} = As + e$, where $e$ is a random noise vector sampled at random over a fixed range of values. The LWE problem is how to obtain the vector $s$ from the matrix $A$ and the vector $\hat{b}$ and the error $e$. Based on LWE of asymmetric key generation algorithms, vector $s$ is used as the private key and matrix $A$ and vector $\hat{b}$ are used as the public key. Due to the addition of random noise, it is not possible to use Gaussian elimination. The only way to find $s$ is by brute force cracking, trying the possible values of $s$ one by one. No quantum algorithm can currently crack this problem, and it is therefore quantum computationally secure.

## 4 Proposed scheme

### 4.1 Framework

Fig 3 shows the core architecture of the private key security management scheme proposed in this article. The relevant research content is mainly divided into four parts:

1. **Private key security generation**. We use the LWE algorithm based on lattice theory to generate public-private key pair.

2. **Generation of a private key encryption key**. We use the user's biometrics to generate a stable and unique key, and use a secret sharing scheme based on the CRT to achieve secret distribution and secret reconstruction of private keys, then fuse the biometric keys to achieve reliable encryption of private keys.

3. **Distributed storage for encrypted private keys**. We use a blockchain to store encrypted private keys, which ensures the consistency of storage based on the blockchain's own mechanism and realizes secure and practical storage.

4. **Recovery of private key**. If the private key is lost, recover the private key by querying and comparing.

The whole process is as follows: firstly, the asymmetric public and private keys are generated by LWE, the multi-biometric key is generated based on the face and fingerprint features

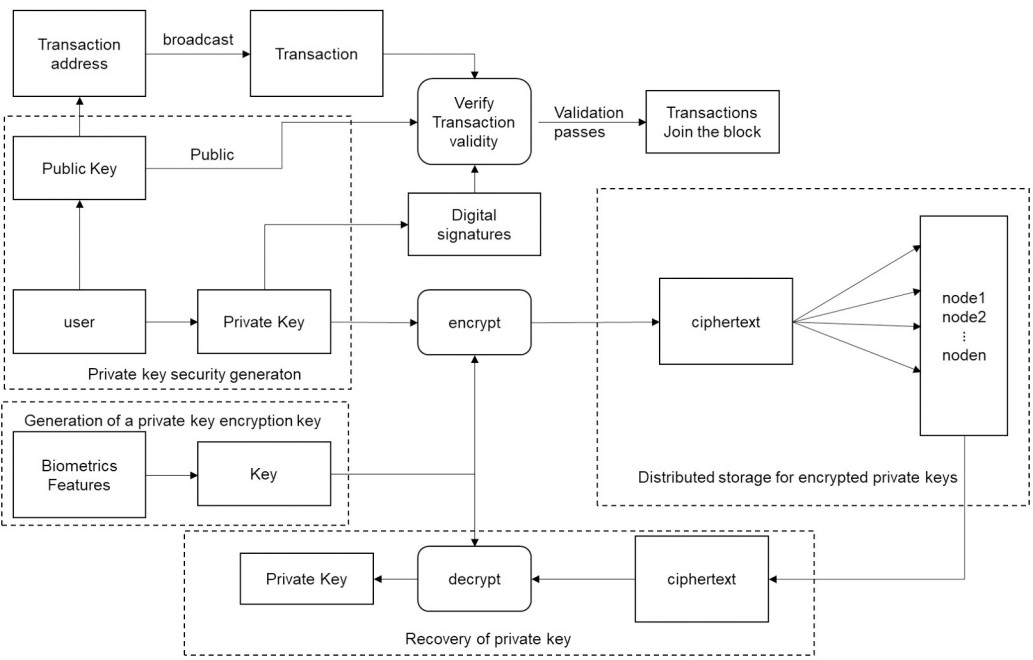

**Fig 3. Private key security management scheme.**

based on the fuzzy extractor, then, the private key is reliably encrypted based on the CRT and biometric key to generate secret fragments, and the secret fragments are stored on the block-chain. If the private key needs to be recovered, the new biometric feature is extracted as input to decrypt the ciphertext to recover the private key.

## 4.2 Private key security generation

In order to resist quantum computing attacks, we use the LWE problem based on lattice theory to ensure the quantum attack resistance of the blockchain system, and generates a public private key pair based on the LWE public key cryptography scheme as shown in Alg 1. In the LWE-based asymmetric key generation algorithm, $n$ is the input system security parameter; the private key is a random number $s$ generated based on system noise and time; $q$ is a prime number between $n^2$ and $2n^2$, all operations are carried out in $Z_q$, that is, all are modulo $q$ operations; $m$ is the number of polynomials; $\alpha$ is the noise parameter; $\chi$ is the probability distribution on $Z_q$ with $\alpha$ as the parameter. Select $m$ linearly independent vector $a_1, \cdots, a_m \in Z_q^n$ from the uniform distribution, select $m$ elements $e_1, \cdots e_m \in Z_q$ according to $\chi$, finally calculate the public key $(a_i, b_i)_{i=1}^m$, and output the public private key pair $\{s, (a_i, b_i)_{i=1}^m\}$.

**Algorithm 1**: LWE-based Asymmetric Key Generation Algorithm

**Input:** security parameter $n$
1 define private key $s$;
2 $q$ = a prime number between $n^2$ and $2n^2$;
3 $m = (1 + \varepsilon)(n + 1)\log q,\ \varepsilon > 0$;
4 $\alpha = \frac{1}{\sqrt{n}\log^2 n}$;
5 **for** $i = 1;\ i \leq m$ **do**
6 $a_i \in Z_q^n$;
7 $e_i \in Z_q$;
8 public key $= (a_i, b_i)_{i=1}^m$;

**Output:** public-private key pair $\{s,(a_i,b_i)_{i=1}^m\}$

We use the LWE-based asymmetric key generation algorithm to generate the public-private key pair of the blockchain $(sk_1, pk_1)$, where the private key $sk_1$ is used to encrypt the hash value of the transaction, and the public key $pk_1$ is used to generate the transaction address. The private key $sk_1$ is used to sign the transaction, and the public key $pk_1$ is used to disclose to each node to verify the validity of the transaction. Users only need a digital signature and the signer's public key to verify the authenticity and integrity of the data. Therefore, the transaction owner can sign the transaction using their private key. Other nodes in the network are then able to verify owner and transaction integrity using only the transaction sender's public key.

## 4.3 Generation of a private key encryption key

In order to realize the reliable generation of the encryption key of the blockchain private key, we propose a stable key generation method based on biometric features. First, the face and fingerprint features are extracted, and then to generate stable and distinguishable descriptors based on feature points, in order to eliminate errors, a stable key generation method based on fuzzy extraction algorithm is proposed to realize key generation based on multiple biometrics. We use the CRT to ensure the security of the private key, finally, the private key is reliably encrypted based on the Chinese remainder theorem and multiple biometrics.

**4.3.1 Stable distinguishable descriptor generation for private key encryption keys.** In order to solve the problem of stable distinguishable descriptor generation for private key encryption keys, we propose a universal and stable distinguishable descriptor extraction method based on the pixel coordinates of two-dimensional feature points. Fig 4 depicts the flow of our method, We first extract the biometric points of faces and fingerprints and calculate their pixel coordinates in the two-dimensional image coordinate system; secondly, to reduce the error of the generated stable distinguishable descriptors, we use the inherent properties of faces and fingerprints to screen out the unstable feature points and keep the stable extracted feature points, and we calculate the stable distinguishable descriptors based on the extracted feature points that are universal at any image resolution to ensure that stable and distinguishable descriptors can be generated for the biometric data collected from the same organism at different times.

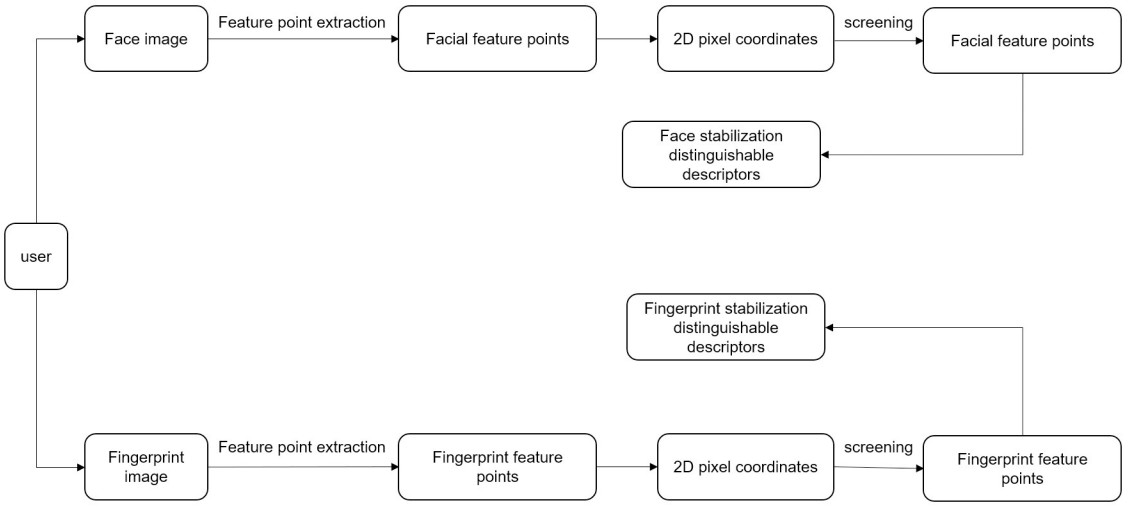

**Fig 4. Stable distinguishable description of the sub-extraction process.**

We define the $n \times 2$ matrix *boldsymbol* $P_{n \times 2}$ to represent the biometric feature points extracted from the image, where each row of the matrix corresponds to the two-dimensional pixel coordinates of a single feature point. The pixel coordinates of the $k$-th feature point are denoted as *boldsymbol* $p_k$. This definition is illustrated in Eq (2).

$$P_{n \times 2} = (p_1, \cdots, p_n)^T \tag{2}$$

To describe $n$ 2D feature point pixel coordinates, this paper employs three control points, where each feature point's true 2D coordinate, *boldsymbol* $p_i$ can be derived by applying a weight to the 2D coordinates of the three control points, $C_j$, as shown in Eq (3). The weighting factor, *boldsymbol* $\alpha_{ij}$, determines the contribution of each control point in determining the true coordinate of the feature point.

$$p_i = \sum_{j=1}^{3} \alpha_{ij} C_j, \text{ with } \sum_{j=1}^{3} \alpha_{ij} = 1 \tag{3}$$

Expanding Eq (3) into matrix form:

$$\begin{bmatrix} p_i \\ 1 \end{bmatrix} = C \begin{bmatrix} \alpha_{i1} \\ \alpha_{i2} \\ \alpha_{i3} \end{bmatrix} = \begin{bmatrix} C_1 & C_2 & C_3 \\ 1 & 1 & 1 \end{bmatrix} \begin{bmatrix} \alpha_{i1} \\ \alpha_{i2} \\ \alpha_{i3} \end{bmatrix} \tag{4}$$

The weighting factor $\alpha_{ij}$ for a given organism's $i$-th feature point $p_i$ is unique when the 2D coordinates of the three control points $C_j$, $j = 1, 2, 3$ are determined which are not collinear. As a result, the weighting factor *boldsymbol* $\alpha_{ij}$ can serve as a universal, stable, and distinguishable biometric descriptor.

The crucial step is to obtain three non-collinear 2D control points from the $n$ 2D pixel feature points. To ensure that the control points are not collinear, we use the theorem that eigenvectors corresponding to different eigenvalues of a matrix are not aligned. The first step is to find the control point $C_1$ using Eq (5), and then construct a new matrix $A$ using Eq (6).

$$C_1 = \frac{1}{n} \sum_{i=1}^{n} p_i \tag{5}$$

$$A = \begin{bmatrix} p_1^T - C_1^T \\ \cdots \\ p_n^T - C_1^T \end{bmatrix} \tag{6}$$

The matrix $A^T A$ is invertible, and therefore its eigenvalues and eigenvectors $\lambda_i$, $i = 1, 2$ can be easily computed. Using the eigenvector decomposition theorem, which states that eigenvectors corresponding to distinct eigenvalues of a matrix are orthogonal, we can find the remaining two control points, $C_2$, $C_3$, as shown in Eq (7).

$$C_2 = C_1 + \lambda_1^{\frac{1}{2}} v_1, C_3 = C_1 + \lambda_2^{\frac{1}{2}} v_2 \tag{7}$$

It can be observed that the control points $C_1$, $C_2$, $C_3$ obtained from the previous steps are non-collinear, thus ensuring that the weighting factor *boldsymbol* $\alpha_{ij}$ is unique for the same organism. Therefore, for a given feature point $p_i$, its corresponding weighting factor

*boldsymbolα*$_{ij}$ can be computed using Eq (8).

$$\begin{bmatrix} \alpha_{i1} \\ \alpha_{i2} \\ \alpha_{i3} \end{bmatrix} = C^{-1} \begin{bmatrix} p_i \\ 1 \end{bmatrix} \tag{8}$$

The stable distinguishable descriptor extraction algorithm is shown in the Alg 2. Fig 5 obtain the face feature points (left) and the fingerprint feature points (right).

**Algorithm 2**: Stable distinguishable descriptor extraction algorithm

```
Input: Feature point coordinates Pnx2 = (p1, ···, pn)T
1 define control points C = {c1, c2, c3};
2 define weighted coefficient αij, with ∑³j=1 αij = 1;
3 define pi = ∑³j=1 αijCj and C1 = 1/n ∑ⁿi=1 pi;
4 define λ = (λ1, λ2) as matrix AᵀA Eigenvalues;
5 define V = (v1, v2) as matrix AᵀA Eigenvectors;
6 C2 = C1 + λ1^½v1, C3 = C1 + λ2^½v2;
7 get weight factors αij = C⁻¹ Pnx2;
8 convert weight factors boldsymbolαij to binary descriptor w;
Output: binary descriptor w
```

Using the algorithm described in this paper, the three non-collinear control points $C_1$, $C_2$, $C_3$ can be obtained based on the facial feature points and fingerprint feature points under the pixel coordinate system. The distribution of these control points and feature points is illustrated in Fig 6. To determine the weighting factor *boldsymbolα*$_{ij}$ for a given feature point $p_i$, the chi-square linear equation described in Eq (8) is solved using the two-dimensional pixel coordinates of the control points and feature points. The resulting binary form of the unique weighted coefficient *boldsymbolα*$_{ij}$ is then used as the final descriptor.

**4.3.2 Multiple biometric key generation.** After creating a stable distinguishable descriptor using the face and fingerprint features as input, there will be an error within a certain range, each calculation result won't be exactly the same, there will be a certain deviation, and it cannot be used as the key directly. Additionally, the key should be made up of random and uniform bits in order to ensure the security of the encryption algorithm. Therefore, we

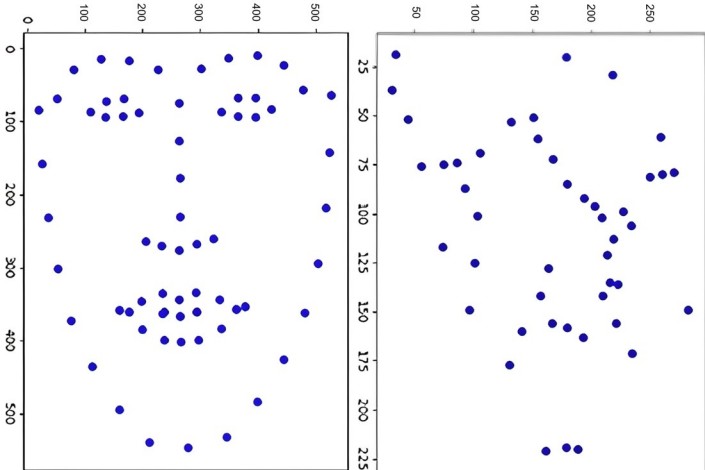

**Fig 5. Distribution of pixel coordinates of facial (left) and fingerprint (right) feature points.**

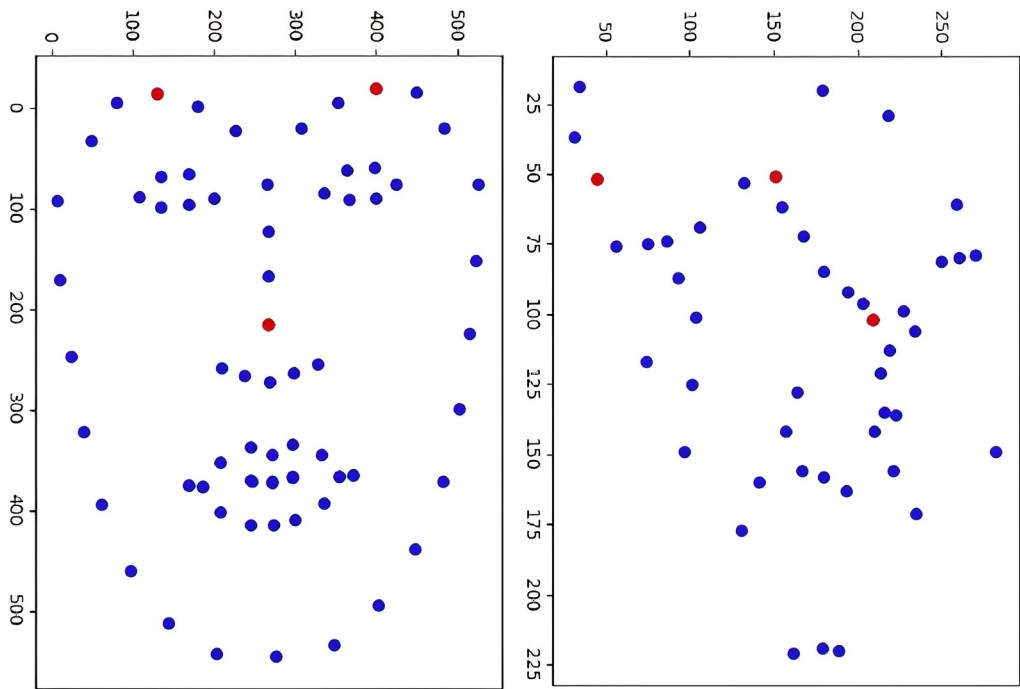

**Fig 6. Distribution of feature and control points, with feature points in blue and calculated control points in red.**

introduce a fuzzy extraction algorithm after biometrics generate stable and distinguishable descriptors, and then generate a stable and distinguishable biological key after the fuzzy extractor in order to produce a key that can be accurately regenerated and is uniformly random.

Use the stable discriminative descriptor as the input random source $w = W_1, W_2, \cdots, W_n$ (in this paper, a string of length n and value range {0, 1} for the generation algorithm of the reusable fuzzy extractor. First choose a random number $R$ as the output of the fuzzy extractor, then randomly sample $w$ to form subset $v_1 = W_{j1}, W_{j2}, \cdots, W_{jk}$ and then use $v_1$ to create a digital locker to hide $R$. Repeat this process to produce $L$ number lockers, which all contain $R$, each of which can be unlocked with $v_1, v_2, \cdots, v_L$ respectively. The generation algorithm uses the face descriptor $D_{fa}$ and fingerprint descriptor $D_{fp}$ calculated based on the stable distinguishable descriptor generation method as input, and obtains a random number key and a public helper character for key recovery respectively, namely $(R_{fa}, P_{fa}) \leftarrow Gen(D_{fa})$, $(R_{fp}, P_{fp}) \leftarrow Gen(D_{fp})$ where $R$ is the key that can be used for encryption, and $P$ is the public key that needs to be saved help string. If the above generation algorithm is repeated for many times, multiple sets of $(R_{fa}, P_{fa})$ or $(R_{fp}, P_{fp})$ corresponding to $R$ and $P$ can be generated, which realizes the reusability of the fuzzy extractor.

When the user enters the biometric again, input random source $w\prime = W'_1, W'_2, \cdots W'_n$. First randomly sample $w'$ to form subset $v'_1 = W'_{j1}, W'_{j1}, \cdots W'_{jk}$ and then use $v'_1$ to try to unlock the digital locker to get $R$. Repeat this process $L$ times, using $v'_1, v'_2, \cdots, v'_L$ to unlock respectively, until $R$ is successfully obtained. Then use the newly generated descriptors $D'_{fa}$ and $D'_{fp}$ as input to obtain the respective public help characters $P_{fa}$ and $P_{fp}$ for key recovery, namely $(R'_{fa}) \leftarrow Rep(D'_{fa}, P_{fa})$, $(R'_{fp}) \leftarrow Rep(D'_{fp}, P_{fp})$. When the Hamming distance of $D$ and $D'$ is less than $t$ and the correct $P$ is used, the random number $R$ can be recovered, namely $R'_{fa} = R_{fa}$, $R'_{fp} = R_{fp}$.

**4.3.3 Secret sharing based on chinese remainder theorem.** In order to protect the private key security, we choose to use the secret sharing technology to realize the secret distribution and secret reconstruction of the private key to achieve the protection function, which divides the private key into several secret fragments and stores them in different nodes respectively. In the secret sharing technique, when the share of secret fragments is not enough to reconstruct the secret, no information about the secret can be obtained; when a sufficient number of secret fragments are obtained, the reconstruction of the secret can be completed and the complete secret information can be obtained. In this paper, the $(k, n)$ threshold construction is completed using a secret sharing scheme based on the CRT, including secret distribution and secret reconstruction.

In the secret distribution stage, we enter the secret $S$, parameters $k$ and $n$ to be shared, select the $(m_1, m_2, \cdots m_n) \in Z^+$ hat meets the conditions, and then calculates the secret fragment $sc_1, \cdots, sc_n$ as output. In the secret reconstruction stage, at least enter the secret fragment $sc_1, \cdots, sc_n$, according to at least $k$ $sc_1, \cdots, sc_k$ pairs, we can get the recovery of secret $S$.

The secret distribution algorithm based on the secret sharing scheme splits the private key $sk_1$ into n secret fragments $SK_1, SK_2, .., SK_n$. When the private key is restored, at least $k$ secret fragments are obtained, and the secret is restored based on the secret reconstruction algorithm to regain the private key as shown in Fig 7. We divide the private key into 20 fragments, and 10 or more of them need to be obtained to recover the full private key.

**4.3.4 Biometric-based private key encryption.** To achieve reliable encryption of private keys, we propose a biometric-based private key encryption algorithm, Alg 3 is the biometrically based private key encryption algorithm. Firstly, a random value of facial features $R_{fa}$ is generated based on a fuzzy extractor. Secondly, LWE-based asymmetric key generation algorithm is used to generate a blockchain public-private key pair $(sk_1, pk_1)$, and the key pair $(sk_2, pk_2)$ is generated based on fingerprint features, then the secret sharing-based scheme divides $sk_1$ into $n$ secret fragments $SK_1, SK_2, .., SK_n$. We will encrypt the key pair $SK_1, SK_2, .., SK_n$ generated by fusing biometrics. Salt encryption refers to a method of adding a salt value to the information to be encrypted, thereby enhancing password security. We use the hash value of $R_{fa}$ to salt $n$ secret fragments $SK_1, SK_2, .., SK_n$ respectively, to obtain $n$ salted encrypted ciphertexts $SK_1', SK_2', \cdots, SK_n'$. the salted ciphertext is asymmetrically encrypted using the public key $pk_2$ to obtain the ciphertext $C_1, C_2, \cdots, C_n$.

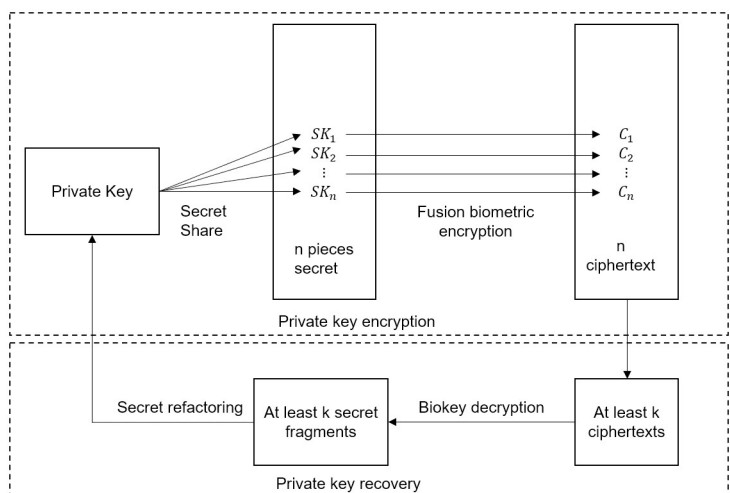

**Fig 7. Private key encryption and private key recovery process.**

**Algorithm 3**: Biometric-based private key encryption algorithm

```
  Input: face key R_fa, fingerprint key R_fp
1 generate (sk₁, pk₁) and (sk₂, pk₂) based on LWE;
2 hash the face key R_fa get hk₁: hk₁ = hash(R_fa);
3 Salt N shares with hk₁: Salt(N(m_i, sc_i));
4 using hash(R_fa) as the private sk₂;
5 generate the corresponding public key use sk₂;
6 use (sk₂, pk₂) to asymmetrically encrypt the N shares: Encrypt{Salt
  (N(m_i, sc_i))};
  Output: N secure key encrypted fragments
```

## 4.4 Distributed storage for encrypted private keys

In view of the cryptographic chain structure of blockchain, distributed nodes and consensus algorithm, we store the secret fragments on the block through multiple transactions to achieve secure and convenient storage of private keys.

The secret fragments generated the ciphertext $C_1, C_2, \cdots, C_n$, package it together with the information needed to restore the secret $sk_1$ and store it as a block transaction. The information that needs to be stored include publicly available information $R_{fa}$ and $R_{fp}$ to help recover $P_{fa}$ and $P_{fp}$ to help authenticate. Constructing $n$ group information $(C_1, P_{fa}, P_{fp}, pk_2), \cdots, (C_n, P_{fa}, P_{fp}, pk_2)$. The information of all blocks in the blockchain is the data that can be seen and shared by all nodes, so in order to prevent attackers from maliciously collecting $n$ group information $(C_1, P_{fa}, P_{fp}, pk_2), \cdots, (C_n, P_{fa}, P_{fp}, pk_2)$ to attack, using $n$ transactions to issue and store in different blocks, while recording the index of the block where the $n$ group information is located and the transaction storage.

## 4.5 Recovery of private key

According to the stored index or query the transaction corresponding to one's own address, at least obtain the information stored in $k$ transactions and verify whether the information has been tampered with, which are recorded as $(C'_1, P_{fa}, P_{fp}, pk_2), \cdots, (C'_n, P_{fa}, P_{fp}, pk_2)$. The steps to recover the private key are as follows:

1. get the ciphertext $C'_1, C'_2, \cdots, C'_n$ to help recover public information $(P_{fa}, P_{fp})$ of $(R_{fa}, R_{fp})$ and key pair the public key $pk_2$.

2. obtain the user's face image and fingerprint image, extract the new face descriptor $D'_{fa}$ and fingerprint descriptor $D'_{fp}$ as input, use the respective public help characters $P_{fa}$ and $P_{fp}$ for key recovery, namely $(R'_{fa}) \leftarrow Rep(D'_{fa}, P_{fa}), (R'_{fp}) \leftarrow Rep(D'_{fp}, P_{fp})$.

3. use $R'_{fa}$ through SHA256 algorithm to get its hash value $hash(R'_{fa})$, and then use the LWE public-private key pair generation algorithm to calculate the key pair $(sk'_2, pk'_2)$, where the private key is $hash(R'_{fa})$. Compare the generated public key $pk'_2$ with $pk_2$ to confirm that the generated public and private key pair is correct, namely $(sk'_2, pk'_2) = sk'_2, pk'_2$. Finally, the ideal private key $sk'_2$ is obtained.

4. use the private key $sk'_2$ to decrypt the obtained ciphertext $C'_1, C'_2, \cdots, C'_k$ to obtain the ciphertext $SK''_1, SK''_2, \cdots, SK''_n$.

5. use $hash(R'_{fp})$ to decrypt $SK''_1, SK''_2, \cdots, SK''_n$, then we can obtain $SK'_1, SK'_2, \cdots, SK'_k$.

6. the secret reconstruction based on the CRT uses $k$ fragments $SK'_1, SK'_2, \cdots, SK'_k$ to reconstruct the secret, that is, the private key $sk'_1$

7. use the LWE public-private key pair generation algorithm to calculate the key pair $(sk'_1, pk'_1)$, where the private key is $sk'_1$. Compare the generated public key $pk'_1$ with $pk_1$ to confirm that the generated public and private key pair is correct, that is $(sk'_1, pk'_1) = (sk_1, pk_1)$. The final successful recovery of the user's private key is $sk_1$.

# 5 Security and performance analysis

## 5.1 security analysis

In this section, we briefly analyze the security of the authentication schemes of the proposed framework, satisfying the following performance and security properties.

**5.1.1 Anonymity.** The proposed scheme ensures the user's biometric data is not stored on any server. Only the stable and distinguishable descriptor generated from the user's biometric feature is used, which is converted into a random value through the fuzzy extractor. Therefore, the user's original biometric feature cannot be restored from the random value.

**5.1.2 Collusion resistance.** The Asmuth-Bloom threshold scheme based on the Chinese remainder theorem is used for secret sharing. If the secret is distributed into n fragments, it is assumed that at least k fragments are required for secret recovery. Even if $k - 1$ nodes collude with each other, they cannot change the calculation result or achieve recovery of the private key.

**5.1.3 Anti-tampering.** The blockchain's non-tampering characteristics, combined with the sum of all private key fragments stored on the blockchain, make it impossible for attackers to tamper with the protocol after its execution.

**5.1.4 Disaster recovery.** Since the fragments are distributed to different nodes for storage during the transaction, even if the data on k-1 nodes is lost, the user can still recover according to the fragments on the remaining nodes.

**5.1.5 Forward secrecy.** The user generates multiple biological feature points through the face and fingerprint, and generates random values for the feature descriptor through the fuzzy extractor. We use the LWE public-private key pair generation method, hash function, and salt processing to encrypt the private key, ensuring forward secrecy.

**5.1.6 Resisting quantum attacks.** The authentication scheme designed based on the lattice public key cryptographic algorithm can resist quantum attacks since there is currently no algorithm that can solve the lattice problem. Literature has also proven that the LWE problem can resist quantum algorithm attacks. Our proposed authentication scheme is based on the LWE problem, ensuring resistance to quantum attacks.

**5.1.7 Resistance to man-in-the-middle attacks.** In our scheme, two-way authentication between users and blockchain nodes is considered. By using asymmetric encryption to store information on blockchain nodes, even if attackers intercept information, they cannot generate keys or tamper with information due to the one-way nature of the hash function and the complexity of LWE, ensuring resistance to man-in-the-middle attacks.

**5.1.8 Resistance to replay attacks.** In the proposed scheme, the salted hash function and LWE are used multiple times in the encryption process to enhance the security of the key fragments. Therefore, even if an attacker intercepts the information transmitted during the authentication process and attempts to carry out replay attacks, they will not be able to obtain any useful information for calculating other related keys through the key. There are generally two scenarios for replay attacks: one is to replay the information transmitted by the device to the server, and the other is to replay the information transmitted by the server to the device. However, due to the robustness of the proposed scheme, both types of replay attacks are effectively prevented.

**Table 1. Security properties comparison.**

| Security properties | [22] | [23] | [16] | [17] | [24] | [25] | [26] | BPKEM |
|---|---|---|---|---|---|---|---|---|
| *anonymity* | Y | Y | Y | Y | Y | Y | Y | Y |
| forward secrecy | Y | Y | Y | Y | N | N | N | Y |
| mutual authentication | Y | Y | Y | Y | Y | Y | Y | Y |
| Resistance to replay attacks | Y | Y | Y | Y | N | N | N | Y |
| tamper proof | Y | Y | Y | Y | Y | Y | Y | Y |
| Resists man-in-the-middle attacks | Y | Y | Y | Y | N | N | Y | Y |
| disaster recovery | N | N | N | Y | N | Y | Y | Y |
| Collusion resistance | N | N | Y | Y | N | Y | Y | Y |
| Use of Biometrics | Y | N | N | N | Y | Y | Y | Y |
| Resistance to biometric errors | Y | N | N | N | Y | Y | N | Y |

We also compare this work with related work shown in Table 1, which confirms the good performance of the proposed method in terms of security properties. We use "Y" to denote the scheme satisfies the property; otherwise it is denoted by "N".

## 5.2 performance analysis

**5.2.1 storage cost.** The information to be stored includes fragmented encrypted ciphertext $C$, $P_{fa}$, $P_{fp}$ are the public information to help restore $R_{fa}$, $R_{fp}$ respectively, and the public key $pk_2$ is used to help with identity verification, the number of ciphertext fragments is set to $n$. Then the total cost of storing a complete ciphertext is: $C + n \times (P_{fa} + P_{fp} + pk_2)$

**5.2.2 Computing costs.** $T_L$ represents the time-consuming calculation of the LWE public-private key pair generation algorithm, $T_H$ represents the time-consuming of hash function encryption and decryption, $T_N$ represents the time-consuming to divide the ciphertext into $n$ pieces using the Chinese remainder theorem, $T_K$ represents the time spent on secret recovery of $k$ shards, $T_S$ represents the time consumption of salt treatment, $T_F$ represents the time-consuming to find $k$ fragments on the blockchain for secret recovery. The total cost calculated is:
$T_L + T_N + T_H + n \times T_S + T_H + T_L + n \times T_L + T_F + T_H + T_L + k \times T_S + T_H + k \times T_L + T_K + T_L = 4 + n + kT_L + T_N + 4T_H + (n + k)T_S + T_F + T_K$.

## 6 Experimental results

In this section, we will conduct experiments to verify the reliability and feasibility of the proposed method. Our experiments are intended to address the following questions:

**RQ1**: How to select suitable feature points to ensure sufficient stability and distinguishability?

**RQ2**: The success rate of the stable discriminant descriptor fuzzy Extraction.

**RQ3**: Feasibility of private key encryption based on BKG and Chinese Remainder Theorem secret sharing algorithm.

The characteristics of the personal computer that was used to perform several tests, i.e., AMD Ryzen 7 5800X 8-Core Processor, RAM Memory of 16 GB in Ubuntu 18.04 environment.

## 6.1 Data set description

The dataset about the facial features is the face recognition dataset of Extended Yale Face Database B [27]. This dataset collects images of the same face in different lighting environments,

which fully meets the data type required for this project. We selected the data of four groups of faces (B11, B12, B13, and B14) under different lighting conditions for experiments.

We conduct experimental verification based on the SOCOFing fingerprint dataset [28]. SOCOFing consists of fingerprint images of multiple ethnicities, including the original fingerprint data and the rotated fingerprint data.

## 6.2 Metrics

**6.2.1 Hamming distance.** The Hamming distance indicates the number of different digits/letters in a pair of messages. In order to calculate the Hamming distance between two strings, and, we perform their XOR operation, $xor = a \otimes b$, and then count the total number of 1s in the resultant string.

**6.2.2 Success rate.** Success rate is the fraction or percentage of success among a number of attempts to perform a procedure or task. In our experiments, Success Rate (SR) is defined as Eq (9):

$$SR = \frac{number\ of\ successes}{total\ number\ of\ experiments} \tag{9}$$

**6.2.3 Average Hamming distance.** Average Hamming distance is the average value of the Hamming distance obtained by randomly sampling $n$ comparisons, Average Hamming distance (AH) is defined as Eq (10):

$$AH = \frac{1}{n} \times \sum_{i=1}^{n} The\ Hamming\ distance\ obtained\ from\ the\ i-th\ sampling \tag{10}$$

**6.2.4 Decline rate.** Decline rate is the percentage between the decline after executing the program or task and the original value. In our experiments, Decline Rate (DR) is defined as Eq (11):

$$DR = \frac{AH\ before\ screening\ feature\ points - AH\ after\ screening\ feature\ points}{AH\ before\ screening\ feature\ points} \tag{11}$$

## 6.3 Experiment on stable discriminative descriptors based on biometrics (RQ1)

**6.3.1 Experiment on stable discriminative descriptors based on facial features.** Three steps are primarily involved in the extraction of stable and distinguishable descriptors for facial features. First, using the HOG directional gradient histogram algorithm, the face image is extracted from the input original image; next, using the Dlib library [29], the specific 82 feature points are extracted from the face image, the distribution of these feature points and the number of them are shown in Table 2; finally, the feature points are classified and chosen; The stable distinguishable descriptor is then determined by categorizing the feature points that have been chosen.

According to the stable and distinguishable description of the sub-extraction process, the face is extracted by the HOG algorithm based on the direction gradient histogram, as shown in Table 3, in the four sets of data selected in this paper, the accuracy of face detection based on

**Table 2. Face feature point types.**

| feature point types | number |
|---|---|
| Chin | 4 |
| Left eyebrow | 12 |
| Right eyebrow | 13 |
| Nose bridge | 11 |
| Tip of the nose | 8 |
| Left eye | 8 |
| Right eye | 9 |
| Upper lip | 7 |
| Lower lip | 10 |

the HOG algorithm has reached more than 80%, and the successful detection of the face image is shown in Fig 8.

After obtaining the successfully detected face image, based on the Dlib library, extract the feature points of the stable distribution of the face, as shown in Fig 9, the horizontal and vertical coordinates respectively represent the coordinate axes based on image pixels after feature points are extracted. With the change of the face expression, the feature points near the eyes and lips will fluctuate greatly, resulting in a large change in the descriptor error. In view of this feature, we screen different types of feature points on the face. Fig 10 is the two-dimensional feature points extracted after screening the feature points based on Fig 9a and 9b, the outer contour of the face and the more obvious and prominent nose part are basically preserved, and the feature points extracted near the eyes and lips are filtered out.

For different biometric acquisition devices, the biometric images obtained by them have great differences in image size, angle, relative distance, and the like. However, for the same creature, its biological features are relatively unchanged in an image of any resolution, and the biological features can be described uniformly without the image size. Therefore, this paper is based on control points to obtain any image with a fixed dimension. The stable distinguishable descriptor is shown in Alg 2. For the same face, the error between descriptors should be within the acceptable range, and for different faces, the error should be much larger than the acceptable range. The descriptors extracted in this paper are binary descriptors, so the Hamming distance is used to calculate the error between the descriptors.

When the key is generated and regenerated based on the stable and distinguishable descriptors of the same creature at different times, there will be a certain probability of wrong regeneration according to the difference of its Hamming distance. It is assumed that the Hamming distance extracted twice is less than or equal to $t$, that is, $dis(w, w') \leq t$. For each $i$, the probability of $v_i' = v_i$ is at least $\left(1 - \frac{t}{n-k}\right)^k$. The probability that the regeneration algorithm does not find a matching $v_i'$ resulting in an output error prompt is $\left(1 - \left(1 - \frac{t}{n-k}\right)^k\right)^L$. In this paper, the

**Table 3. Face detection success rate of HOG algorithm.**

| Dataset | Data volume | Accurate number | Failure number | SR |
|---|---|---|---|---|
| $B11$ | 64 | 52 | 12 | 81.25% |
| $B12$ | 64 | 56 | 8 | 87.5% |
| $B13$ | 64 | 52 | 2 | 81.25% |
| $B14$ | 64 | 53 | 11 | 82.81% |

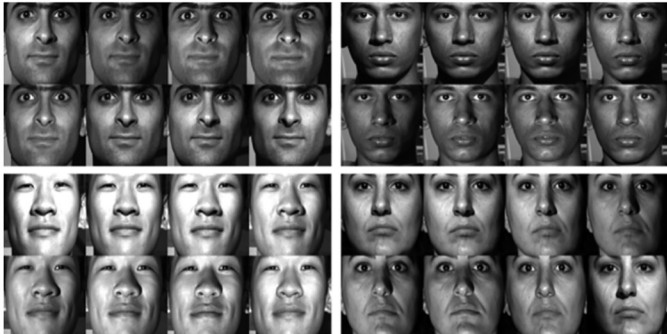

**Fig 8. Face image extracted based on HOG algorithm.**

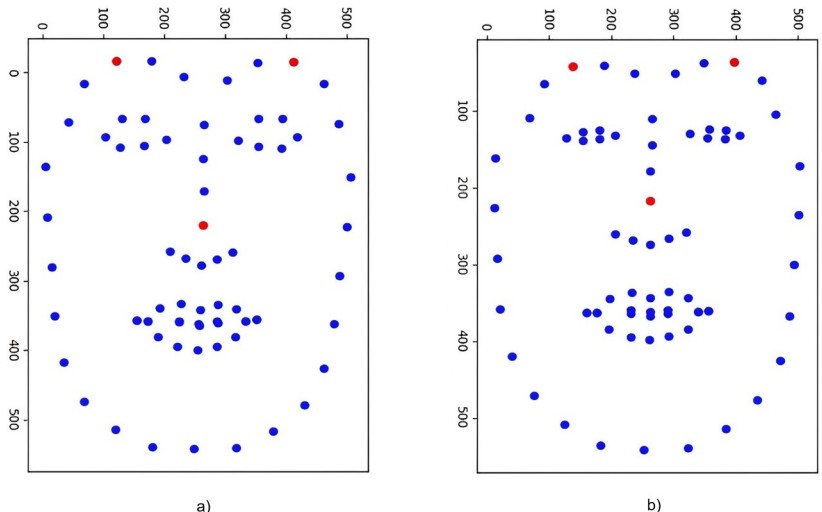

**Fig 9. Distribution of all feature points.** a) and b) represent the changes of feature points under different expressions of the same person.

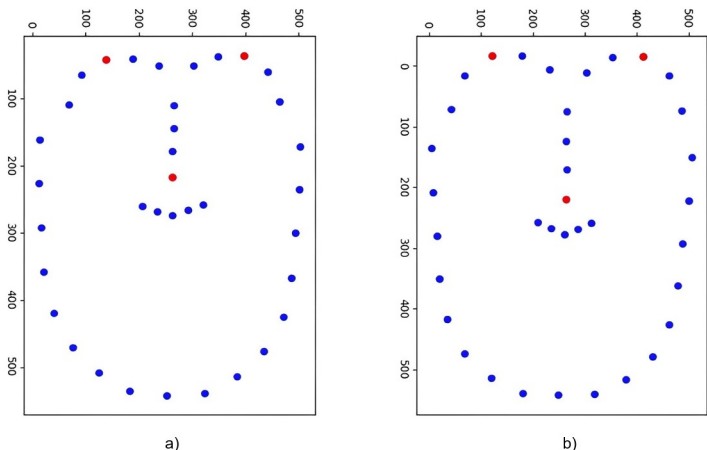

**Fig 10. Characteristic point distribution after screening.**

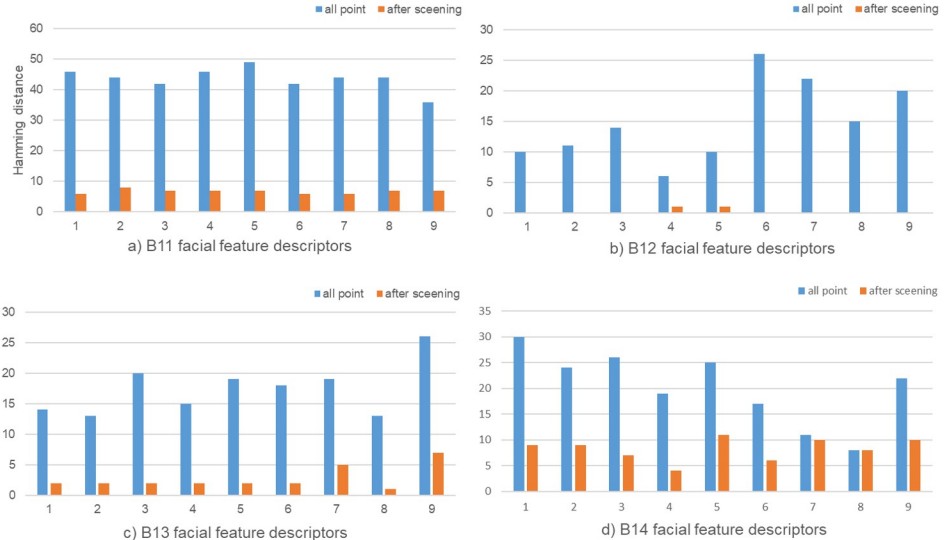

**Fig 11. Hamming distance error before and after screening feature points.**

input $n$ is 128 bits, $t = 10$ bits are allowed to be different, the output unique key $k = 16$ bits. When the sampling times $L = 20$, according to the above formula, the key error regeneration probability is $6.28e^{-4}$, when the sampling times $L = 100$, the key error regeneration probability is $9.773e^{-12}$.

The descriptor calculated in this paper is 128 bits and can be obtained with a 99.99% probability when the Hamming distance is within 10 based on the fuzzy extractor's basic principle. The descriptor is taken from the facial feature points in Fig 9; however, stability is not met because of the great hamming distance. To fully meet the error range necessary for fuzzy extraction, extract the descriptor from the landmarks depicted in Fig 10. We randomly sampled 9 times, and calculated the Hamming distance of the descriptor before and after feature point screening, as shown in Fig 11 and Table 4. It is clear that before screening the feature points, the Hamming distance between the describers is extremely unstable, even up to 30 bits of error, and after screening the feature points, the Hamming distance error is reduced to less than 10. According to the calculation and comparison of the data in the figure, it can be seen that the average Hamming distance difference of the four groups of descriptors before and after removing the feature points is 36.89, 14.67, 14.66, 12.00, respectively, and the decline rate is 84.47%, 98.52%, 84.06%, 59.36%. Satisfy the requirements of fuzzy extraction to generate a unique key. Fig 11 and Table 4 show the stability of the descriptors obtained by screening feature points. This paper also analyzes the distinguishability of different faces.

In this paper, the Hamming distance of the descriptors is calculated in two pairs of four sets of face data, Fig 12 and Table 5 shows the Hamming distance error of the descriptor between

**Table 4. Hamming distance error before and after screening feature points.**

| Dataset | AD before screening | AD after screening | decline | DR |
|---|---|---|---|---|
| $B11$ | 43.67 | 6.78 | 36.89 | 84.47% |
| $B12$ | 14.89 | 0.22 | 14.67 | 98.52% |
| $B13$ | 17.44 | 2.78 | 14.66 | 84.06% |
| $B14$ | 20.22 | 8.22 | 12.00 | 59.36% |

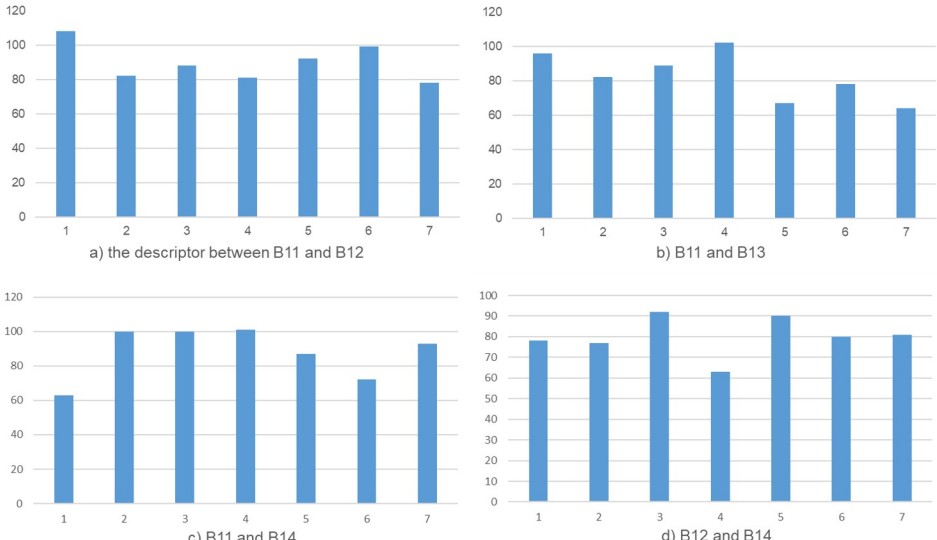

**Fig 12. Hamming distance of the descriptor between four sets of experimental data.**

B11 and B12, B11 and B13, B11 and B14, B12 and B14. According to the calculation of the data in the figure, it can be known that the average Hamming distance of the four groups of descriptors between different person is 89.71, 82.57, 88, and 80.14 respectively, which are obviously much greater than 10.

From the above experimental analysis, it can be seen that the descriptor error between different face data is much larger than the threshold. Therefore, the proposed stable and distinguishable descriptor based on control points not only has sufficient stability for the same face data in different periods, but also has distinguishability between different faces.

**6.3.2 Experiment on stable discriminative descriptors based on fingerprint features.**
To improve the quality of the raw fingerprint data, it is necessary to enhance the dataset using an appropriate algorithm. The fingerprint enhancement algorithm exploits the well-defined frequency properties of the fingerprint image, targeting locally stable frequency variations in the valleys and ridges of the fingerprint image. In order to achieve this, Gabor filters are used to stabilize the tuning to the appropriate frequency and direction. This results in the removal of noise between valleys and ridges, while retaining a clear ridge-valley structure. Fig 13 provides a visual representation of the enhanced fingerprint image.

The corresponding feature points are extracted from the enhanced fingerprint image based on the pixel characteristics of the ridge end and fork points. Fig 14 shows the fingerprints of 6 different people. The enhanced fingerprint image has a very fine structure. The figure illustrates how feature point extraction is extremely unstable only around the fingerprint image due to edge issues. The feature point at the edge of the fingerprint image must be removed

**Table 5. Hamming distance of the descriptor between four sets of experimental data.**

| AD | B11 | B12 | B13 | B14 |
|---|---|---|---|---|
| *B*11 | 0 | 89.71 | 82.57 | 88 |
| *B*12 | 89.71 | 0 | 87.71 | 80.14 |
| *B*13 | 82.57 | 87.71 | 0 | 87.71 |
| *B*14 | 88 | 80.14 | 87.71 | 0 |

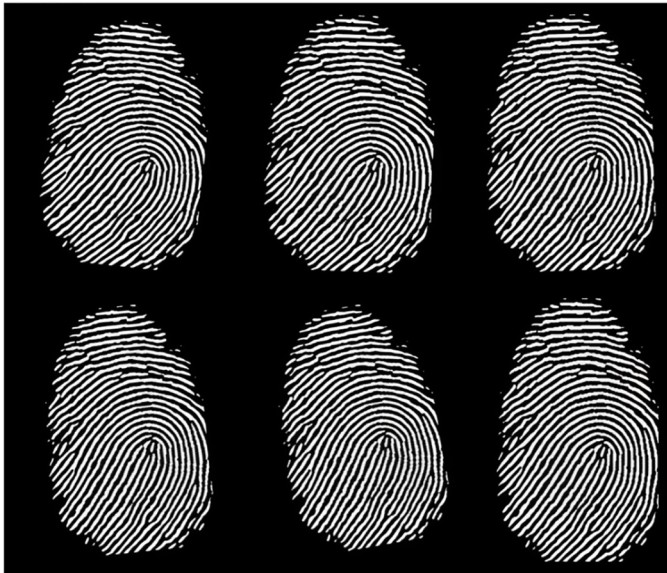

**Fig 13. Enhanced fingerprint image.** fingerprints of 6 different people.

because, in the case of normal acquisition, the fingerprint image typically does not contain structural mutations. We calculate the stable and distinguishable descriptors of the fingerprint image based on Alg 2, and analyze the errors of the fingerprint descriptors before and after screening feature points.

We randomly sampled 7 times, and calculated the Hamming distance of the descriptor, Fig 15 and Table 6 illustrates how the Hamming distance error between the profilers, A, B, C, D

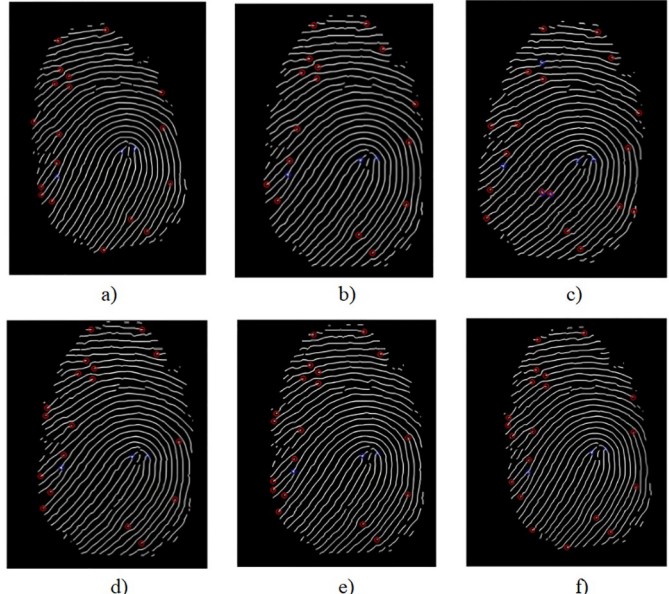

**Fig 14. Fingerprint image feature point extraction results.** fingerprints of 6 different people.

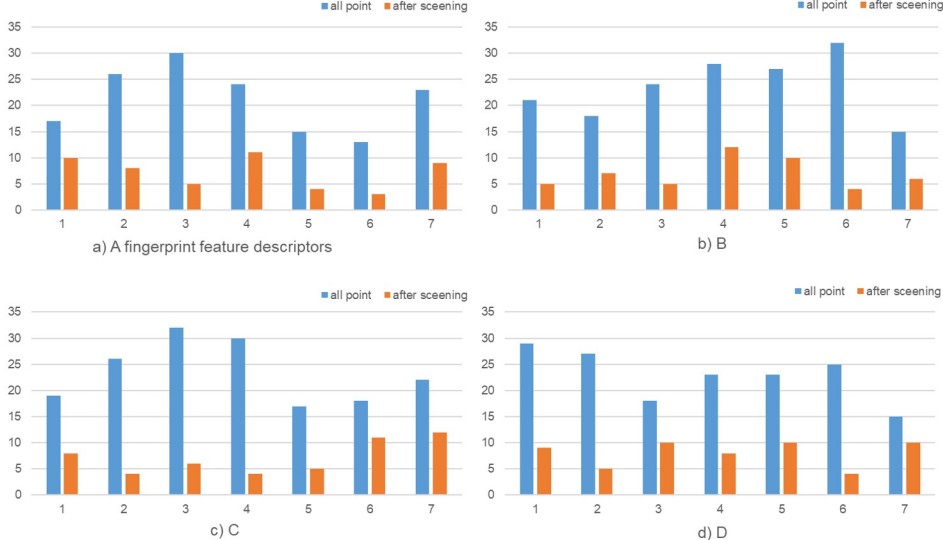

**Fig 15. Fingerprint description sub-Hamming distance before and after screening.**

represent four fingerprints from different people. Before screening out the edge feature points, the Hamming distance error between descriptors can reach 30, but after screening, it remains within 10. According to the calculation of the data in the figure, it can be known that the average Hamming distance difference of the four groups of fingerprint feature descriptors before and after removing the feature points is 14.00, 16.57, 16.29, and 14.86 respectively, and the decline rate is 66.23%, 70.30%, 69.53%, 65.00%. Achieving the stability of the same biometric fingerprint feature descriptor.

As shown in Fig 16 and Table 7, A, B, C, D represent four fingerprints from different people. the stable distinguishable distance between various biometric profiles is well beyond the error range. According to the calculation of the data in the figure, it can be known that the average Hamming distance of the four groups of descriptors between different person is 74.00, 81.29, 81.00, and 77.57 respectively. This distance, known as the Hamming distance, fully satisfies the requirement for distinguishability between fingerprint profilers.

## 6.4 Stable discrimination descriptor fuzzy extraction experiment (RQ2)

From the aforementioned experiments, it is possible to derive, respectively, 128-bit binary descriptors based on face data and 128-bit binary descriptors based on fingerprint data. However, the calculated descriptors are still used even when the filtered feature points are used. It is necessary to remove the error based on a fuzzy extractor and obtain a distinct biometric key because there is still a Hamming distance error within 10 of the target value. We examined the

**Table 6. Fingerprint description sub-Hamming distance before and after screening feature points.**

| Dataset | AD before screening | AD after screening | decline | DR |
|---------|--------------------|--------------------|---------|-----|
| A | 21.14 | 7.14 | 14.00 | 66.23% |
| B | 23.57 | 7.00 | 16.57 | 70.30% |
| C | 23.43 | 7.14 | 16.29 | 69.53% |
| D | 22.86 | 8.00 | 14.86 | 65.00% |

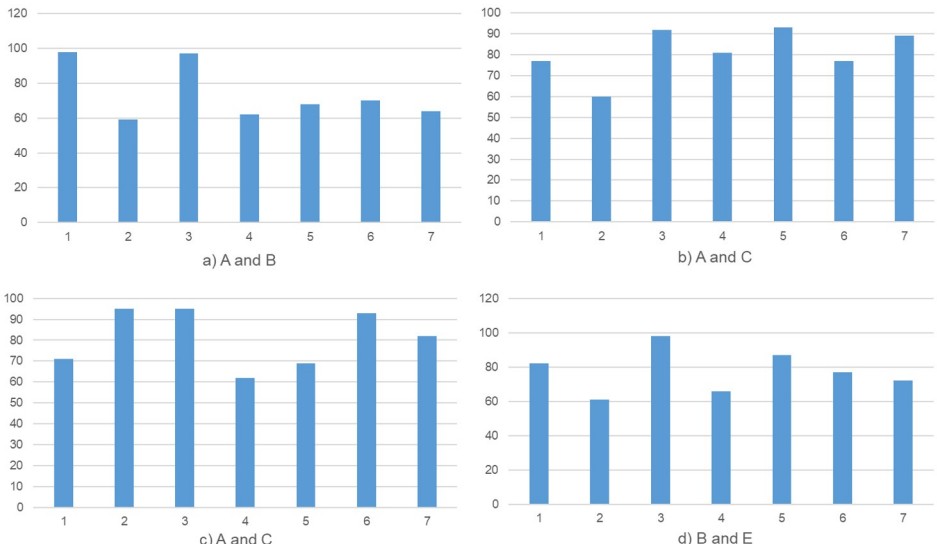

**Fig 16. The sub-Hamming distance between different biometric fingerprints.**

**Table 7. The sub-Hamming distance between different biometric fingerprints.**

| Dataset | AD |
|---------|-----|
| A | 74.00 |
| B | 81.29 |
| C | 81.00 |
| D | 77.57 |

fuzzy extraction's rate of success. The success rate of face descriptor fuzzy regeneration is displayed in Table 8. It can be seen that in 1000 experiments, the success rate is almost 100% when the Hamming distance error is less than 10.

Similar to the previous experiment, we also run the same test on fingerprint descriptors and achieve 100% regeneration power when the Hamming distance error is within 10. As a result, it has been established that both face data and fingerprint data can produce distinctive keys.

## 6.5 Private key sharing and encryption experiment based on BKG and CRT (RQ3)

Ethereum is an open-source blockchain platform that enables developers to build blockchain-based applications. In this article, we constructed a private blockchain system using the

**Table 8. Face descriptor fuzzy extraction experiment.**

| Dataset | Experiment number | Accurate number | Failure number | SR |
|---------|-------------------|-----------------|----------------|------|
| B11 | 1000 | 1000 | 0 | 100% |
| B12 | 1000 | 1000 | 0 | 100% |
| B13 | 1000 | 1000 | 0 | 100% |
| B14 | 1000 | 1000 | 0 | 100% |

**Table 9. Equipment used in the experiment.**

| module | version |
|---|---|
| CPU | AMD Ryzen 7 5800X 8-Core Processor |
| RAM | 16GB |
| system | Ubuntu18.04 |

Ethereum Geth platform and conducted experiments to verify the proposed private key encryption and decryption algorithm based on biometrics. Table 9 shows the equipment used in the experiment, and we set the parameters of the Ethereum creation block as indicated in Table 10. To facilitate effective testing of the algorithm, we set transaction fees to zero and reduced the hash calculation difficulty required for block mining.

As shown in Fig 17, this article finally built a command-line Ethereum blockchain.

The public-private key pair is generated based on the LWE algorithm as shown in Table 11. According to experiment 1, $t = 10$, if we choose $L = 20$ based on the condition that the success rate is similar between 20 and 200 times, the time consumed can be reduced. The generated private key is secretly shared based on the CRT, as shown in Table 12, we share the private key into 20 secret fragments, and set the minimum recovery private key fragmentation threshold to 10.

The unique key of face data is used to salt 20 private key fragments, and then the unique key of fingerprint data is used as a new private key $sk_2$, as shown in Table 11. For the public key $pk_2$ paired with the fingerprint and private key, use $(sk_2, pk_2)$ to encrypt the $N$ fragments generated by the private key $sk_1$ to obtain the private key fragments encrypted with salt, as shown in Table 13.

After salted encryption, this article obtains the confidential fragment and distributes it over the blockchain node. When restoring the private key, only 10 or more of the 20 fragments must be obtained from the blockchain node. The fragments of the private key are then desalted and decrypted using the updated face and fingerprint data, and processed using the CRT to produce the final recovered private key, as shown in Table 14. If there are only 9 fragments, the recovered private key is wholly incorrect. The private key can be fully recovered once there are 10 copies of the fragments.

**Table 10. The Ethereum genesis block parameters.**

| parameter | value |
|---|---|
| nonce | 0x0000000000000042 |
| Timestamp | 0x0 |
| parenHash | 0x0000000000000000000000000000000000 |
| extraData | 0x00 |
| gaslimit | 0x8000000 |
| difficulty | 0x400 |
| mixhash | 0x0000000000000000000000000000000000<br>0000000000000000000000000000000000 |
| coinbase | 0x3333333333333333333333333333333333333333 |
| config | "chainId": 987, "homesteadBlock": 0,<br>"eip155Block": 0, "eip158Block": 0 |

**Fig 17. Schematic diagram of a private blockchain system based on Ethereum.**

**Table 11. Public-private key pair generated baesd on LWE.**

| Type of keys | Key content |
|---|---|
| private key $sk_1$ | 0b1111110101010100 |
| public key $pk_1$ | 75,52,96,87,69,$\cdots$, 50,17,19,1,51,31 <br> 33,5,45,24,82, $\cdots$, 88,81,95,57,44,89 |
| Fingerprint unique key corresponds to private key $sk_2$ | 0b1100110001111111111110 |
| Fingerprint unique key corresponds to public key $pk_2$ | 20,24,17,61,09,$\cdots$, 54, 24, 11, 2, 04, 19 <br> 95,30,35,99,19,$\cdots$, 93, 53, 33, 43, 56, 02 |

# 7 Conclusion and future work

It is especially crucial to ensure the security of the private key because, in blockchain technology, the security of the user's assets and the security of the entire blockchain system are directly impacted by the security of the private key. For the whole process of private key generation, encryption, storage and recovery, this paper proposes a biometric-based private key encryption and management framework for blockchain (BPKEM), and conducts experimental verification.

The security of the entire generation, encryption, storage, and recovery of private keys are discussed, we propose a biometric-based private key management framework (BPMF). In order to achieve the secure generation of private keys against quantum computing, the

**Table 12. CRT-Based secret fragment.**

| Fragment number | Fragment content |
|---|---|
| M_0 | 0b100000011001000010101000000000 <br> 0101010001101011000110100000101 |
| 0_S_M | 0b100001011101000011001100011 <br> 1111111111001010001010111010101000 <br> 0b100001001011111100010010101110 <br> 1011110001111001010001110100001 |
| . . . | . . . |
| 19_S_M | 0b100110101001001110001011101 <br> 1100010011100000000111100111101 <br> 0b111111111101101100110011111101 <br> 0111111010101000010111100110011 |

**Table 13. Encrypt and salt the secret shard.**

| Fragment number | Fragment content |
|---|---|
| M_0 | b"\x04\xe9\xee\d=\xb4V)y\x8e\x12\xd\...7\xadl\x11\xf5\x9ae \xfbln!" |
| 0_S_M | b'\x04W\x18\x04=\xe2\xe5"9\x04K\x08\x00\xe7\x97\...\x94\xfb\xa7' |
| ... | ... |
| 19_S_M | b'\x04\xd9\x91\xbdH\xee\xf8\xea\xe...:\xe1\xd4c\xda\xcf3\x9d \xf5' |

blockchain system first uses the LWE-based asymmetric key generation algorithm to generate public and private key pairs. Then, the secret distribution and encryption of the private key is achieved by using the secret sharing and biometric key of the Chinese residual theorem. After encryption, a reliable storage of the private fragments is implemented based on the blockchain system itself. Finally, a reliable private key recovery scheme is realized, and systematic experiments are carried out to talk about the capability of private key regeneration and recovery.

For the private key encryption problem, due to the need for conventional keys that face problems such as storage security, a biometric-based stable key generation (BKG) method is proposed, using the key generated based on multiple biometrics to encrypt the private key. The bio-key generation uses the bio-key generation method based on stable and distinguishable descriptors in this paper. The key generation of face features and fingerprint features are experimented with and discussed in this article. The stable distinguishable descriptor is primarily divided into three steps. First, we obtain the face and fingerprint images from the relevant dataset, extract the biometric points of the face and fingerprint, and calculate their pixel coordinates in the two-dimensional image coordinate system; secondly, in order to reduce the error of the generated stable distinguishable descriptors, the unstable feature points are screened out and the stable extracted feature points are retained to calculate the stable distinguishable descriptors using the inherent properties of faces and fingerprints; finally, for the errors caused by the angle and resolution of face and fingerprint images collected at different periods, a control point-based stable distinguishable descriptor extraction algorithm is proposed using the invariance of the relative distribution of face and fingerprint feature points, and the errors of the stable distinguishable descriptors extracted from the biometric data collected for the same organism at different periods are controlled within a certain range. The success rate of the fuzzy extractor to regenerate the key is discussed by using the descriptor as the input of the fuzzy extractor.

On the basis of our work, the possible follow-up improvements and possible research directions are summarized as follows: Firstly, in terms of biometric key generation, the method based on deep learning can be used to further improve and strengthen. Secondly, for the private key security management scheme, in which distributed storage is based on blockchain,

**Table 14. CRT-Based private key recovery.**

| private key | 0b1111110101010100 |
|---|---|
| 9 fragments restore the private key | 0b1000011111110100110011010 1100100011010100010101011000 |
| 10 fragments restore the private key | 0b1111110101010100 |

can explore the possibility of proposing identity authentication-based transaction storage and acquisition schemes based on smart contracts or other methods.

## Author Contributions

**Conceptualization:** Hao Cai, Han Li, Jianlong Xu, Yue Zhang.

**Data curation:** Han Li, Linfeng Li.

**Formal analysis:** Han Li.

**Funding acquisition:** Hao Cai, Jianlong Xu.

**Investigation:** Hao Cai, Jianlong Xu.

**Methodology:** Han Li, Jianlong Xu, Yue Zhang.

**Project administration:** Hao Cai, Linfeng Li.

**Resources:** Hao Cai, Jianlong Xu.

**Software:** Han Li.

**Supervision:** Hao Cai, Jianlong Xu, Linfeng Li.

**Validation:** Hao Cai.

**Visualization:** Han Li, Yue Zhang.

**Writing – original draft:** Hao Cai, Han Li, Jianlong Xu.

**Writing – review & editing:** Han Li, Jianlong Xu.

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
