## [Decision Letter · Decision Letter 0]

20 Feb 2023

PONE-D-22-34006BPKEM: a biometric-based private key encryption and management framework for blockchainPLOS ONE

Dear Dr. Xu,

Thank you for submitting your manuscript to PLOS ONE. After careful consideration, we feel that it has merit but does not fully meet PLOS ONE’s publication criteria as it currently stands. Therefore, we invite you to submit a revised version of the manuscript that addresses the points raised during the review process.

We look forward to receiving your revised manuscript.

Kind regards,

Pandi Vijayakumar, Ph.D

Academic Editor

PLOS ONE

Journal Requirements:

"We gratefully acknowledges 2021 Guangdong province special fund for science and technology project, Grant Number: STKJ2021201; Research on Food Production and Marketing traceability Software system based on Blockchain, Grant Number: STKJ2021011; 2020 Li Ka Shing Foundation Cross-Disciplinary Research Grant, Grant Number: 2020LKSFG08D; Guangdong basic and applied basic research fund project, Grant Number: 2021A1515012527; Free application project of Guangdong Natural Science Foundation, Grant Number: 2018A030313438; Special projects in key fields of colleges and universities in Guangdong Province, Grant Number: 2020ZDZX3073,2022ZDZX1008. "

"Funding: This research was financially supported by 2021 Guangdong province special fund for science and technology project, Grant Number: STKJ2021201; 

Research on Food Production and Marketing traceability Software system based on Blockchain, Grant Number: STKJ2021011; 

2020 Li Ka Shing Foundation Cross-Disciplinary Research Grant, Grant Number: 2020LKSFG08D; 

Guangdong basic and applied basic research fund project, Grant Number: 2021A1515012527;

Free application project of Guangdong Natural Science Foundation, Grant Number: 2018A030313438; 

Special projects in key fields of colleges and universities in Guangdong Province, Grant Number: 2020ZDZX3073,2022ZDZX1008.

5. We note that Figure 5 includes an image of a [patient / participant / in the study]. 

Reviewers' comments:

Reviewer's Responses to Questions

**Comments to the Author**

1. Is the manuscript technically sound, and do the data support the conclusions?

Reviewer #1: Yes

Reviewer #2: Yes

2. Has the statistical analysis been performed appropriately and rigorously? 

Reviewer #1: Yes

Reviewer #2: N/A

3. Have the authors made all data underlying the findings in their manuscript fully available?

Reviewer #1: Yes

Reviewer #2: Yes

4. Is the manuscript presented in an intelligible fashion and written in standard English?

Reviewer #1: Yes

Reviewer #2: Yes

5. Review Comments to the Author

Reviewer #1: In this paper, the authors propose a biometric-based stable key generation (BKG) method to encrypt the blockchain’s private key and guarantee the security of the entire process of creating private keys, encrypting them, and recovering them.

What’s more, the contributions of this paper can be summarized into two aspects including proposing a comprehensive management solution and creating distinguishing descriptors based on several biometric factors.

The whole paper is finished with some other problems. There is a long way to go before it can be accepted. We point out the specific problems as follows:

1. There is no performance analysis in this paper. Both security and efficiency are the necessary indices to assess the value of a work. However, no performance analysis is found in this paper. As a consequence, the superiority of the proposed protocol in performance is not demonstrated.

2. In Related work, the authors don’t explain clearly why they use the main technology they have presented. What’s more, the authors just simply show the references’ main idea and they don’t analyze these references and explain the development of this main technology.

3. In section 4.3.1, the authors propose a universal and stable distinguishable descriptor extraction method. However, they don’t give a specific explanation. And the given schematics can’t explain well about the methodology. There are many other similar methods similar to what this paper shows. So, the authors should give a detailed and clear explanation, not a few similar words.

4. There are some typos in this paper. For example, in contributions, the sentence “we propose A comprehensive management solution for the blockchain’s private key system …...” should be “we propose a comprehensive management solution for the blockchain’s private key system …...” Please carefully check this paper and correct similar errors.

5. Table and pictures in this paper are not standard. It is suggested that the author check the format of the full text to improve its readability of the full text.

Reviewer #2: A novel method based on bio-metrics is introduced for private key encryption

Section 3 must be strengthened

section 5 must include more experimental graphical results

some of the following interesting works on block chain must be reviewed

1. "A secure and efficient authentication and data sharing scheme for Internet of Things based on blockchain."

2. "BBAAS: Blockchain-Based Anonymous Authentication Scheme for Providing Secure Communication in VANETs."

3. "EAAP: Efficient Anonymous Authentication With Conditional Privacy-Preserving Scheme for Vehicular Ad Hoc Networks."

6. PLOS authors have the option to publish the peer review history of their article (what does this mean?). If published, this will include your full peer review and any attached files.

Reviewer #1: No

Reviewer #2: No

---

## [Author Response · Author response to Decision Letter 0]

14 Apr 2023

Dear Sir/Madam,

Thank you very much for all your comments. We made substantial changes in response to your comments, and fell the quality of this paper had a great improvement as a result of updating this paper taking your suggestions. Thank you very much for your valuable comments and please check our updates in response to each comment:

Response to academic editor:

Point 1: Please ensure that your manuscript meets PLOS ONE's style requirements, including those for file naming. 

Response 1: Thanks for your suggestion. We double-checked the format of the manuscript and modified the format of relevant figures and tables to meet PLOS ONE's style requirements.

Point 2: Please note that PLOS ONE has specific guidelines on code sharing for submissions in which author­generated code underpins the findings in the manuscript. In these cases, all author-generated code must be made available without restrictions upon publication of the work.

Response 2: Thanks for your suggestion. We shared the code about biometric feature extraction and generating stable and distinguishable descriptors on github, and the access link of the code is: https://github.com/liiihan/BPKEM. Since the size of the blockchain part exceeds the shareable size limit, it cannot be shared on GitHub. If you want to get unlimited codes for codes, please contact us by email: xujianlong@stu.edu.cn

Point 3: We note that you have provided funding information that is not currently declared in your Funding Statement. However, funding information should not appear in the Acknowledgments section or other areas of your manuscript. We will only publish funding information present in the Funding Statement section of the online submission form. Please remove any funding-related text from the manuscript and let us know how you would like to update your Funding Statement.

Response 3: Thanks for your suggestion. We have removed any funding-related text from the manuscript.

Point 4: In your Data Availability statement, you have not specified where the minimal data set underlying the results described in your manuscript can be found. PLOS defines a study's minimal data set as the underlying data used to reach the conclusions drawn in the manuscript and any additional data required to replicate the reported study findings in their entirety. All PLOS journals require that the minimal data set be made fully available.

Response 4: Thanks for your suggestion. The datasets we use are all public datasets and are completely open. We have added relevant links about the public datasets we used in the revised cover letter.

Point 5: We note that Figure 5 includes an image of a patient participant in the study. If you are unable to obtain consent from the subject of the photograph, you will need to remove the figure and any other textual identifying information or case descriptions for this individual.

Response 5: Thanks for your suggestion. We have checked the relevant pictures, the pictures used are the pictures in the public data set, and the publisher has authorized.

Response to reviewer 1:

Point 1: There is no performance analysis in this paper. Both security and efficiency are the necessary indices to assess the value of a work. However, no performance analysis is found in this paper. As a consequence, the superiority of the proposed protocol in performance is not demonstrated.

Response 1: Thanks for your suggestion. We have added section 5 to analyze the security and performance of the scheme. We briefly analyze the security of the authentication scheme of the proposed framework and compare it with other schemes, and calculate the storage cost and computation cost of the scheme

The changes are reflected in lines 432 to 500.

Point 2: In Related work, the authors don't explain clearly why they use the main technology they have presented. What's more, the authors just simply show the references' main idea and they don't analyze these references and explain the development of this main technology.

Response 2: Thanks for your suggestion. We have supplemented some of the analysis of references in the related work, and also added supplements to the paper on the private key authentication phase, and explained the reasons for proposing BPKEM at the end of the section.

The changes are reflected in lines 73 to 152.

Point 3: In section 4.3.1, the authors propose a universal and stable distinguishable descriptor extraction method. However, they don't give a specific explanation. And the given schematics can't explain well about the methodology. There are many other similar methods similar to what this paper shows. So, the authors should give a detailed and clear explanation, not a few similar words.

Response 3: Thanks for your suggestion. We have supplemented the specific explanation of the stable and distinguishable descriptor extraction method, by extracting feature points to generate feature point coordinates in the two-dimensional image coordinate system, and then obtaining stable and distinguishable descriptors by calculating the two-dimensional control points of the feature points. We have reworked the relevant flow charts to make the description of the differentiable descriptor extraction method clearer and easier to understand

The changes are reflected in lines 268 to 319.

Point 4: There are some typos in this paper. For example, in contributions, the sentence “we propose A comprehensive management solution for the blockchain's private key system ...…“ should be "we propose a comprehensive management solution for the blockchain's private key system ..…. “ Please carefully check this paper and correct similar errors.

Response 4: Thanks for your suggestion. We double checked the paper and corrected some typos in the paper.

Point 5: Table and pictures in this paper are not standard. It is suggested that the author check the format of the full text to improve its readability of the full text.

Response 5: Thanks for your suggestion. We have checked the format of the full text, and modified non-standard diagrams to improve the readability of the full text.

Response to reviewer 2:

Point 1: Section 3 must be strengthened.

Response 1: Thanks for your suggestion. We supplemented the concept and principle of the fuzzy extractor in Chapter 3, and added an explanation about LWE.

The changes are reflected in lines 154 to 214.

Point 2: section 5 must include more experimental graphical results.

Response 2: Thanks for your suggestion. We added some related charts in Section 6, and added two metrics, Average Hamming distance and Decline rate. In Experiment 1, we added a description table of face feature extraction points and the experimental data table of experiment on stable discriminative descriptors based on facial features. Added Supplementary instructions on fingerprint enhancement and the experimental data table of experiment on stable discriminative descriptors based on fingerprint features. In Experiment 3, the equipment, Ethereum creation block parameter table and private chain system diagram are supplemented.

The changes are reflected in lines 532 to 676.

Point 3: some of the following interesting works on block chain must be reviewed

1. "A secure and efficient authentication and data sharing scheme for Internet of Things based on blockchain."

2. "BBAAS: Blockchain-Based Anonymous Authentication Scheme for Providing Secure Communication in VANETs."

3. "EAAP: Efficient Anonymous Authentication With Conditional Privacy-Preserving Scheme for Vehicular Ad Hoc Networks."

Response 3: Thanks for your suggestion. We reviewed papers 1 and 2 in the related work in Section 2, and compared the corresponding schemes of 1, 2 and 3 in the security performance analysis of Table 1.

The changes are reflected in lines 122 to 136 and Table 1.

---

## [Decision Letter · Decision Letter 1]

9 May 2023

BPKEM: a biometric-based private key encryption and management framework for blockchain

PONE-D-22-34006R1

Dear Dr. Xu,

We’re pleased to inform you that your manuscript has been judged scientifically suitable for publication and will be formally accepted for publication once it meets all outstanding technical requirements.

Kind regards,

Pandi Vijayakumar, Ph.D

Academic Editor

PLOS ONE

Additional Editor Comments (optional):

The authors have done all the corrections given in the previous round and hence the paper can be accepted in the present format.

Reviewers' comments:

Reviewer's Responses to Questions

**Comments to the Author**

1. If the authors have adequately addressed your comments raised in a previous round of review and you feel that this manuscript is now acceptable for publication, you may indicate that here to bypass the “Comments to the Author” section, enter your conflict of interest statement in the “Confidential to Editor” section, and submit your "Accept" recommendation.

Reviewer #2: All comments have been addressed

2. Is the manuscript technically sound, and do the data support the conclusions?

Reviewer #2: Yes

3. Has the statistical analysis been performed appropriately and rigorously? 

Reviewer #2: N/A

4. Have the authors made all data underlying the findings in their manuscript fully available?

Reviewer #2: Yes

5. Is the manuscript presented in an intelligible fashion and written in standard English?

Reviewer #2: Yes

6. Review Comments to the Author

Reviewer #2: All the comments given by the reviewer are done.

Hence, the manuscript can be accepted.

This manuscript can be published in the broad subject area.

7. PLOS authors have the option to publish the peer review history of their article (what does this mean?). If published, this will include your full peer review and any attached files.

Reviewer #2: No

---

## [Editor Report · Acceptance letter]

22 Jun 2023

PONE-D-22-34006R1 

BPKEM: a biometric-based private key encryption and management framework for blockchain 

Dear Dr. Xu:

I'm pleased to inform you that your manuscript has been deemed suitable for publication in PLOS ONE. Congratulations! Your manuscript is now with our production department. 

Kind regards, 

on behalf of

Dr. Pandi Vijayakumar 

Academic Editor

PLOS ONE